# WEASEL: Out-of-Domain Generalization for Web Agents via Importance-Diversity Data Selection

**Fatemeh Pesaran Zadeh** [1] [*]   **Seyeon Choi** [1] [*]   **Xing Han Lù** [2] [3]   **Siva Reddy** [2] [3] [4]   **Gunhee Kim** [1]

## Abstract

Large language models (LLMs) have enabled web agents that follow natural language goals through multi-step browser interactions. However, agents fine-tuned on specific trajectories and domain often struggle to generalize out of domain, and offline training can be compute-inefficient due to noisy, redundant trajectories and long accessibility-tree (AXTree) states. To address both issues, we propose WEASEL, a trajectory selection method for offline training of web agents. WEASEL selects a fixed-budget subset of trajectory steps by optimizing an objective that balances unary importance with pairwise diversity over states, websites, and interaction patterns, solving efficiently with a greedy algorithm. We further improve efficiency with target-centered AXTree pruning that keeps only content around the ground-truth action target, and we mitigate style mismatch for reasoning-native models by replacing expert traces with model-generated, style-consistent rationales. Across AgentTrek and NNetNav training datasets, evaluations in WebArena, WorkArena, and MiniWob, and experiments with Qwen2.5-7B, Gemma3-4B, and Qwen3-8B, WEASEL improves out-of-domain performance while reducing training cost, producing roughly 9.7-12.5× training speedups over standard fine-tuning. We make the code available at https://github.com/fatemehpesaran310/weasel.

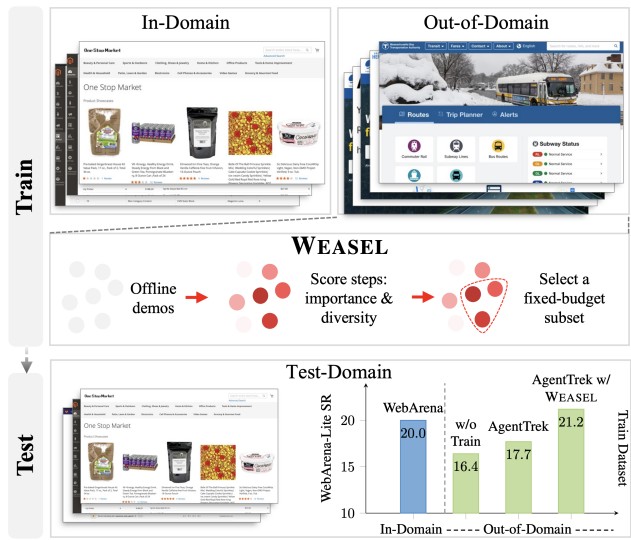

*Figure 1.* Overview of WEASEL. Conventional trained web agents show a sharp performance drop under out-of-domain shifts to unseen websites and interaction patterns. WEASEL tackles this challenge via novel trajectory selection: it scores offline demonstration steps for goal relevance and diversity, then applies greedy subset selection under a fixed budget. Agents trained with WEASEL generalize better to unseen test domains, achieving a +4.8 gain in zero-shot transfer from AgentTrek (Xu et al., 2024)→WebArena-Lite (Qi et al., 2024) with Qwen3-8B (Team, 2025). Note that bar-top labels indicate the training data source.

## 1. Introduction

Large Language Models (LLMs) have moved beyond text generation to support multi-step reasoning, planning, and action in interactive environments (Wang et al., 2023; Liu et al., 2023). This progress has accelerated interest in LLM-based web agents that map natural-language goals to sequences of browsing (Qi et al., 2024; Zhou et al., 2024a). Recent improvements are fueled by strong foundation models and large-scale instruction data (Ou et al., 2024; Murty et al., 2025; Xu et al., 2025b), along with agent-specific training such as imitation learning and reinforcement learning (Qi et al., 2025; Wei et al., 2025), enabling agents to solve increasingly complex, long-horizon web tasks.

Despite this progress, recent web agents remain limited in two key respects. First, agents fine-tuned on specific

---
[*]Equal contribution  [1]Seoul National University  [2]McGill University  [3]Mila – Quebec AI Institute  [4]Canada CIFAR AI Chair. Correspondence to: Gunhee Kim <gunhee@snu.ac.kr>.

*Proceedings of the 43rd International Conference on Machine Learning*, Seoul, South Korea. PMLR 306, 2026. Copyright 2026 by the author(s).

trajectories and environments often fail to generalize to out-of-domain tasks and settings. Performance often drops substantially when deployed on the websites, layouts, or interaction patterns that differ from those seen during training (Murty et al., 2025). Recent studies mostly evaluate under in-benchmark settings (Qi et al., 2024; Lai et al., 2024; Zhou et al., 2024b; Shen et al., 2025; Wei et al., 2025), training on a subset of tasks from a benchmark environment and testing on held-out tasks within the same environment (e.g., WebArena-to-WebArena in Figure 1). This limits their applicability in real-world web tasks where distribution shifts are inevitable.

Second, offline web interaction data are often noisy and redundant (Nekoei et al., 2025; Xu et al., 2025a); expert trajectories can be overly long while containing relatively sparse task-relevant signal, due to redundant steps, drifting behaviors, or partially misaligned actions. Thus, models are prone to overfitting to narrow patterns rather than learning robust, transferable behaviors. This motivates improving both learning efficiency and generalization ability by curating an informative and diverse subset of trajectories and pruning distracting page content.

To address these two challenges, we propose WEASEL (**WE**b **A**gents with trajectory **SEL**ection), a data selection approach for offline web agent training that improves both learning efficiency and out-of-domain generalization. WEASEL formulates a fixed-budget subset selection problem with a quadratic objective that balances unary importance and pairwise diversity (§2.2). The unary importance prioritizes the steps that are semantically relevant to the goal, while the pairwise diversity encourages coverage over heterogeneous states, websites, and interaction patterns. Since the resulting subset selection problem is NP-hard in general, we employ an efficient greedy algorithm that scales to large collections of long trajectory in practice (§2.3). WEASEL improves (1) out-of-domain generalization by explicitly encouraging diversity, which mitigates overfitting to website-specific artifacts (Figure 1), and (2) training efficiency by removing redundant and noisy steps from long trajectories, achieving higher accuracy with up to $12.5\times$ training speed-up compared to training on the full data (Table 1).

To further improve training efficiency, we introduce target-centered pruning. In web agent settings, each state can contain a large accessibility tree (AXTree), where action-relevant cues may be diluted by substantial irrelevant page content (Deng et al., 2023; Kerboua et al., 2025a). Thus, we prune each state by retaining AXTree content localized around the ground-truth action target, compactly preserving the information most relevant for predicting the expert action (§2.4). This pruning is orthogonally applicable and effectively reduces both the input length and training cost.

For reasoning-native models such as Qwen3 (Yang et al.,

2025), we find that fine-tuning with reasoning traces written by a different model can be detrimental. The issue arises when the reasoning style in the training data differs from that used in the pretraining of the model; as a result, it can harm both out-of-domain generalization and overall performance. To address this, we let the model self-generate style-consistent reasoning traces while preserving the original action supervision in the trajectories (§2.5). We find this step crucial for improving out-of-domain generalization in reasoning-native models (Table 4).

In summary, our contributions are as follows.

- To the best of our knowledge, we propose WEASEL, the first data selection approach for offline web agent training that is explicitly designed to achieve two goals: (1) improved out-of-domain generalization and (2) more compute-efficient learning. WEASEL formulates a subset selection objective that balances unary importance and pairwise diversity. We develop a greedy algorithm that scales to large trajectory collections. We also introduce target-centered AXTree pruning to remove redundant context and accelerate training. For reasoning-native models, we synthesize style-consistent reasoning traces to mitigate style mismatch during fine-tuning.

- We perform a diverse evaluation; web agents trained from two offline datasets, AgentTrek (Xu et al., 2024) and NNetNav (Murty et al., 2025), are evaluated across multiple benchmarks (e.g., WebArena (Zhou et al., 2024a), WorkArena (Drouin et al., 2024a), and Mini-Wob (Shi et al., 2017; Liu et al., 2018)) in a zero-shot manner, using multiple model families (e.g., Qwen2.5-7B (Qwen et al., 2025), Gemma3-4B (Team et al., 2025) and Qwen3-8B (Team, 2025)). We provide ablations that isolate the effects of subset selection, state pruning, and reasoning synthesis. Beyond substantial out-of-domain generalization gains (see Table 1), WEASEL improves training efficiency over standard SFT baselines, achieving 9.7-12.5$\times$ training speed-ups depending on the model.

## 2. WEASEL

WEASEL is a data selection method for training of offline web agents from long, noisy demonstrations. Given an expert trajectory dataset, WEASEL constructs a compact supervision set by selecting a fixed-budget subset of informative yet diverse steps. To further reduce training time, we prune each Web state to retain only action-relevant AXTree content (§2.4). For reasoning-native models, self-reasoning synthesis mitigates reasoning-style mismatch when the training traces are produced by a different model (§2.5).

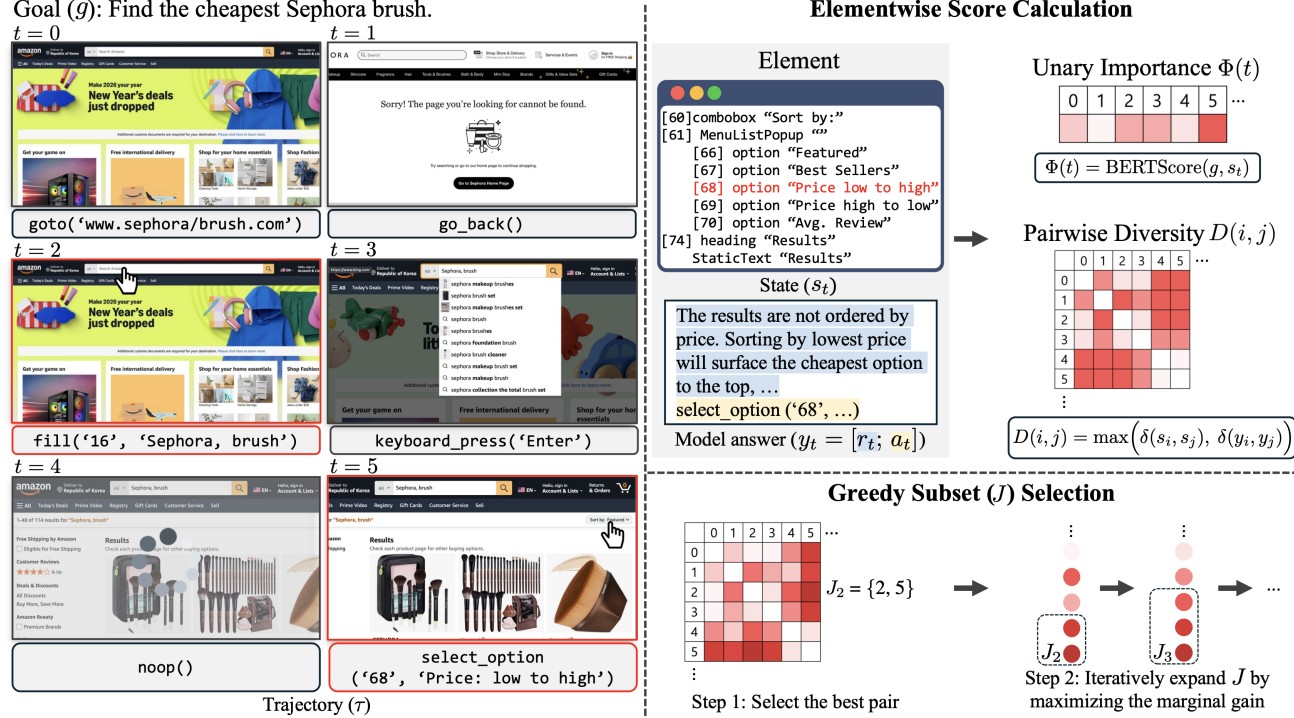

*Figure 2.* (**Left**): An example of a curated trajectory after applying WEASEL. Although the original collected data contain noisy steps ($t = 4$), and erroneous actions ($t = 0$), WEASEL selects a compact subset that retains only the most informative steps (in red) for the goal. (**Right**): Overview of WEASEL. We first perform element-wise score calculation using unary importance and pairwise diversity. WEASEL then applies a greedy subset selection procedure to derive an informative subset $J$. The algorithm initializes $J$ by selecting the best-scoring pair that maximizes the objective, and then iteratively adds the element $k$ that yields the largest marginal gain, until the budget is met (Algorithm 1).

## 2.1. Problem Formulation

**Setup.** Let $\mathcal{D} = \{\tau^{(n)}\}_{n=1}^{N}$ denote an offline dataset of expert web trajectories. Each trajectory $\tau \in \mathcal{D}$ is a sequence $\tau = \left(g, \{(s_t, h_t, a_t, r_t)\}_{t=0}^{T-1}\right)$, where $g$ is a natural-language goal, $s_t$ is a Web state, $h_t$ is an interaction history up to step $t$, $a_t \in \mathcal{A}$ is an expert action (e.g., `click`, `type`, `scroll`, `goto`), and $r_t$ is a reasoning trace. We learn a policy $\pi_\theta$ that imitates the expert in an offline manner by predicting both the action and the reasoning trace at each step, conditioning on the goal, interaction history, and Web state, i.e., $\pi_\theta(a_t, r_t \mid g, h_t, s_t)$. We denote the model answer by $y_t = [r_t; a_t]$, formed by concatenating the reasoning trace and action.

**States and actions for web agents.** We represent each state $s_t$ as a linearization of the page accessibility tree (AXTree). We write the linearized node sequence as $V_t = [v_{t,1}, v_{t,2}, \ldots, v_{t,K_t}]$, where each node $v_{t,k}$ has a unique identifier (e.g., a `bid`), and $K_t = |V_t|$ is the number of nodes in the linearization. For the action space $\mathcal{A}$, we follow BrowserGym (de Chezelles et al., 2025) to include node-grounded actions that reference an AXTree element via its identifier (e.g., `click`, `type`) as well as non-node actions that do not require an identifier (e.g., `goto`, `noop`).

## Algorithm 1 WEASEL subset selection (greedy)

**input** Goal $g$, trajectory $\{(s_t, y_t)\}_{t=0}^{T-1}$, budget $T_0$, weight $\lambda$
1: Compute $\Phi(t) = \text{BERTScore}(g, s_t)$ for all $t$
2: Compute (or cache on-demand) $D(i, j)$ via Eq. (3)
3: $(i_1, i_2) \leftarrow \arg\max_{i<j} \Phi(i) + \Phi(j) + \lambda D(i, j)$
4: $J \leftarrow \{i_1, i_2\}$
5: **for** $m = 3$ to $T_0$ **do**
6:     $i_m \leftarrow \arg\max_{k\notin J} \Phi(k) + \lambda \sum_{i\in J} D(k, i)$
7:     $J \leftarrow J \cup \{i_m\}$
8: **end for**
**output** $J^* \leftarrow \text{sort}(J)$

**Goal.** Since raw web trajectories include many redundant steps and noisy, overly long demonstrations, WEASEL constructs a compact supervision set by selecting a fixed-budget subset of steps $J \subseteq \{0, \ldots, T - 1\}$ with $|J| = T_0 \ll T$. This yields the curated dataset $\tilde{\mathcal{D}} = \bigcup_{\tau \in \mathcal{D}} \{(g, h_t, s_t, a_t, r_t)\}_{t \in J^*(\tau)}$, where $J^*(\tau)$ is obtained by our goal-conditioned importance-diversity objective (§2.2) and solved using a greedy algorithm (§2.3).

## 2.2. Goal-conditioned Subset Selection

Trajectories in the training set for Web agents are often unnecessarily long. In AgentTrek (Xu et al., 2024), a single trajectory contains up to 45 state-action pairs, although only a small fraction is directly relevant to the task goal. Training on long, redundant trajectories is also prone to overfitting to environment-specific patterns. We therefore cast trajectory curation as a subset selection problem that retains only the most informative steps while maintaining diversity among the retained steps to improve generalization.

Given a trajectory of length $T$, we select a compact subset of $T_0 \ll T$ steps; let $J \subset \{0, \dots, T-1\}$ denote the selected indices with $|J| = T_0$. We maximize the objective that combines unary importance and pairwise diversity:

$$\underset{J \subseteq \{0,\dots,T-1\}}{\text{maximize}} \quad \sum_{j \in J} \Phi(j) \;+\; \lambda \sum_{\substack{i < j \\ i,j \in J}} D(i,j) \qquad (1)$$

$$\text{s.t.} \quad |J| = T_0,$$

where $\Phi(j)$ is an *importance* score and $D(i,j)$ is a *diversity* score, and $\lambda$ controls the tradeoff.

For importance $\Phi(j)$, we score each step $s_t$ by its semantic relevance to the goal:

$$\Phi(t) = \text{BERTScore}(g, s_t), \qquad (2)$$

where we use the BERTScore (Zhang et al., 2020) to quantify how well the content of step $t$ aligns with the goal $g$. In principle, BERTScore can be replaced by any embedding similarity; we study alternative similarity models in Section B. However, maximizing importance alone can select redundant, highly similar states and may encourage overfitting to website-specific patterns. We therefore incorporate a pairwise diversity term $D(i,j)$ defined as the maximum pseudo-distance between either the states or the associated model answers:

$$D(i,j) = \max\Big(\delta(s_i, s_j),\ \delta(y_i, y_j)\Big), \qquad (3)$$

where $y_i = [r_i;\, a_i]$ denotes the model answer for step $i$ formed by concatenating its reasoning trace $r_i$ and action $a_i$, and the pseudo-distance $\delta(x, y)$ between two text sequences $x$ and $y$ is defined as

$$\delta(x,y) \triangleq 1 - \text{BERTScore}(x,y), \qquad (4)$$

so that $\delta(x, y)$ is small when $x$ and $y$ are semantically similar. Together, these terms yield the subset selection objective in Eq. (1) under the constraint in set size. Unless otherwise stated, we set $\lambda = 1$ in all experiments.

### 2.3. The Greedy Algorithm

While the problem in Eq. (1) is NP-hard (Garey & Johnson, 2002), it can be formulated as a variant of the max-sum diversification problem, which maximizes a modular

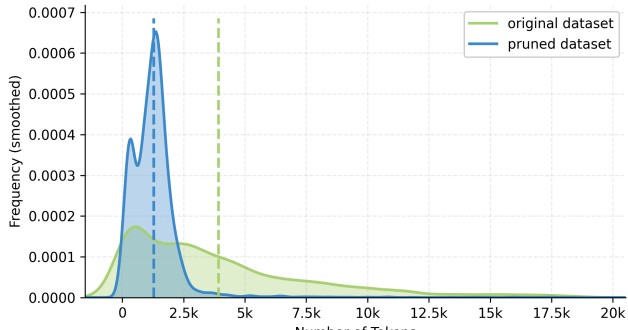

*Figure 3.* Token distribution of 10K subsamples of AgentTrek (Xu et al., 2024) before pruning (green) and after target-centered pruning (blue). Pruning substantially reduces long-tail states, making the resulting sequences more manageable for training.

quality term plus a sum of pairwise distances under a cardinality constraint (Borodin et al., 2017). For metric distances, a greedy algorithm achieves a constant-factor guarantee, including a 2-approximation (Borodin et al., 2017, Theorem 4.1). In our problem, $D(i,j)$ is defined using semantic pseudo-distances (e.g., $1 - \text{BERTScore}$ and a max-composition) that do not satisfy metric properties, so this guarantee does not formally apply. Nevertheless, greedy selection remains an efficient and effective heuristic in practice. We empirically examine the approximation quality of this greedy procedure in §3.3.

We first initialize the selected set by choosing the best pair:

$$(i_1, i_2) = \arg \max_{0 \le i < j < T} \Big(\Phi(i) + \Phi(j) + \lambda D(i,j)\Big),$$
$$J_2 = \{i_1, i_2\}. \qquad (5)$$

Starting from this seed, for $m = 3, 4, \dots, T_0$, we expand the set by adding the index that yields the largest marginal gain with respect to the objective:

$$\Delta(k \mid J_{m-1}) \coloneqq \Phi(k) + \lambda \sum_{i \in J_{m-1}} D(k,i),$$
$$i_m = \arg \max_{k \notin J_{m-1}} \Delta(k \mid J_{m-1}), \qquad (6)$$
$$J_m = J_{m-1} \cup \{i_m\}.$$

We output $J^\star = J_{T_0}$ as the selected subset of indices. The full procedure is summarized in Algorithm 1.

In terms of complexity, computing all unary scores $\Phi(t)$ takes $O(T)$. In addition, precomputing all pairwise distances $D(i,j)$ costs $O(T^2)$ (or they can be cached on demand). Given cached distances, each greedy iteration evaluates marginal gains for up to $T$ candidates, leading to an overall cost of $O(T_0 T)$ for the selection stage.

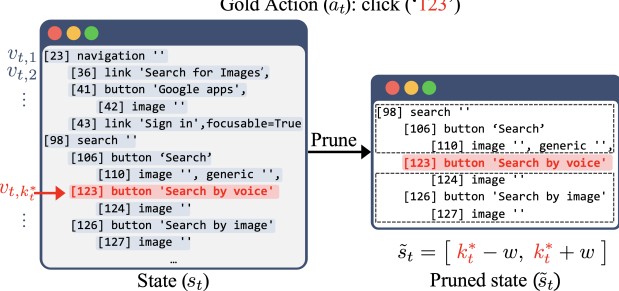

Gold Action ($a_t$): click ('123')

*Figure 4.* An illustration of **Target-centered Pruning**. Given a state $s_t$ in the form of AXTree and gold action $a_t$, we retain only the AXTree elements within a fixed window of size $w$ centered at the target index $k_t^*$, producing the pruned state $\tilde{s}_t$. The $k$-th node in the linearized AXTree at step t is denoted $v_{t,k}$ (e.g., $v_{t,1}$, $v_{t,2}$), and $v_{t,k_t^*}$ is the gold target node.

## 2.4. Target-centered Pruning

Web states can be prohibitively long (Deng et al., 2023; Lù et al., 2024; Kerboua et al., 2025a) because the accessibility tree (AXTree) includes highly repeated and peripheral content (e.g., headers, sidebars, duplicated menus). In AgentTrek, a single linearized state can reach 180K tokens (Xu et al., 2024), far exceeding the context limits of most training and inference setups (Figure 3). This length is not only costly but also noisy; only a small subset of nodes informs the expert's next action. To address this, we propose target-centered pruning, which preserves a compact neighborhood around the target element of the expert action, since local context is shown to be more informative for learning node-grounded web behaviors (Liu et al., 2025).

Specifically, at step $t$, let $V_t = [v_{t,1}, \ldots, v_{t,K_t}]$ be the linearized AXTree node sequence. For node-grounded actions, let $k_t^*$ be the gold target index such that $v_{t,k_t^*} = u_t^*$, where $u_t^* \in V_t$ denotes the corresponding *target node*. We use $(V_t, k_t^*)$ to define target-centered pruning. Given a fixed window size $w \in \mathbb{N}$, we keep a contiguous window of length $(2w+1)$ centered at $k_t^*$. Let

$$\mathcal{K}_t = [\, k_t^* - w, \; k_t^* + w \,] \cap \{1, \ldots, K_t\}. \qquad (7)$$

We retain the subsequence $\tilde{V}_t = [\, v_{t,k} : k \in \mathcal{K}_t \,]$ and denote the pruned (linearized) state as $\tilde{s}_t$ (Figure 4). For non-node actions (e.g., goto, noop), we instead keep a fixed-length prefix under the same length budget (e.g., $2w+1$ or a fixed $L$) to obtain $\tilde{V}_t$. We compare our pruning method against prior state-pruning baselines in Table 3, and further analyze the importance of retaining target-local context in Figure 5. This target-centered pruning is applied as a preprocessing step, prior to running WEASEL.

## 2.5. Self-Reasoning Synthesis for Reasoning Models

Reasoning-native models are fine-tuned to emit intermediate reasoning traces before producing final outputs (DeepSeek-

AI, 2025; Team, 2025). When fine-tuning such models on expert traces $r_t$ produced by a different model or annotation process, we observe that naive fine-tuning can introduce a style mismatch. In practice, this mismatch destabilizes training and may degrade performance, sometimes even falling below the base model before fine-tuning.

To mitigate this, we replace expert reasoning traces with *self-synthesized* traces that match the base model's reasoning style (Zelikman et al., 2022), while keeping the action supervision unchanged. For each selected step $t \in J^*$, we prompt the target reasoning model to generate a trace $\hat{r}_t$ conditioned on the goal, history, pruned state, and the expert action:

$$\hat{r}_t \sim q(\, r \mid g, h_t, \tilde{s}_t, a_t \,), \qquad (8)$$

where $q$ is implemented by prompting the base model to produce a plausible rationale consistent with $a_t$. Refer to Section F.2 for full details of the prompt. We then fine-tune the policy to predict both the synthesized trace and the expert action:

$$\max_\theta \sum_{\tau \in \mathcal{D}} \sum_{t \in J^\star(\tau)} \log \pi_\theta\big(a_t, \hat{r}_t \mid g, h_t, \tilde{s}_t\big). \qquad (9)$$

We ablate the effect of self-reasoning synthesis in Table 4.

## 3. Experiments

### 3.1. Setup

**Baselines.** We compare WEASEL against standard supervised fine-tuning (SFT) baselines under an offline training using AgentTrek (Xu et al., 2024) and NNetNav (Murty et al., 2025) datasets. For LLMs of agents, we use the pretrained Qwen2.5-7B-Instruct (Qwen et al., 2025), Gemma3-4b-IT (Team et al., 2025) and Qwen3-8B (Yang et al., 2025). For all models and datasets, we report performance obtained by reproducing the training process using the hyperparameters specified in Section A. We test baselines that are fine-tuned with different configurations of training sets: (i) fine-tune on the full AgentTrek training set of 52K datapoints with multiple epoch budgets (e.g., 2/4 epochs, depending on the model), (ii) on a uniformly sampled subset of 10K datapoints using the same epoch count as WEASEL, (iii) on a sampled subset via LLM-as-Judge (Zheng et al., 2023), whose details can be found §F.1.

**Evaluation of Out-of-Domain Generalization.** While agents are trained on AgentTrek or NNetNav, we measure their success rates (SR) on WebArena-Lite (Liu et al., 2024), WebArena (Zhou et al., 2024a), MiniWob (Shi et al., 2017; Liu et al., 2018), and WorkArena (Drouin et al., 2024a). For evaluation, we use GPT-4.1-mini as the judge model on WebArena-Lite and GPT-4-1106-preview on WebArena. For consistent benchmark execution, we follow the default

*Table 1.* Success rates (SR) under a zero-shot transfer setting. We fine-tune Qwen2.5-7B-Instruct (Qwen et al., 2025), Gemma3-4b-IT (Team et al., 2025), and Qwen3-8B (Team, 2025) on AgentTrek (Xu et al., 2024), and evaluate on different benchmarks, WebArena-Lite (Liu et al., 2024), WebArena (Zhou et al., 2024a), MiniWob (Shi et al., 2017; Liu et al., 2018), and WorkArena (Drouin et al., 2024b), with no training.

| Model | WebArena-lite SR | WebArena SR | MiniWob SR | WorkArena L1 SR | WorkArena L2 SR | # Data | Training # Epochs | Training Time | Speed-up |
|---|---|---|---|---|---|---|---|---|---|
| *Qwen2.5-7B-Instruct* | 5.5 | 5.2 | 41.8 | 4.8 | 0.0 | – | – | – | – |
| + Full (52K steps) | 10.9 | 8.7 | 44.6 | 12.1 | 0.4 | 52K | 4 | 136.0 hr | 1.0× |
| + Pruning (52K steps) | 9.7 | 8.6 | 47.7 | **12.4** | 0.4 | 52K | 4 | 62.0 hr | 2.2× |
| + Pruning + Sampling (10K steps) | 9.1 | 8.1 | 46.7 | 9.8 | 3.0 | 10K | 4 | 12.0 hr | 11.3× |
| + Pruning + LLM-Judge (10K steps) | 8.5 | 7.8 | 45.4 | 8.5 | 3.0 | 10K | 4 | 12.0 hr | 11.3× |
| **+ WEASEL (10K steps)** | **14.5** | **9.5** | **48.0** | **12.4** | **4.7** | 10K | 4 | 12.0 hr | 11.3× |
| *Gemma3-4B-IT* | 6.7 | 2.7 | 27.4 | 3.6 | 0.0 | – | – | – | – |
| + Full (52K steps) | 9.1 | 4.3 | 28.6 | 3.3 | 0.0 | 52K | 2 | 80.0 hr | 1.0× |
| + Pruning (52K steps) | 9.1 | 4.8 | 28.2 | 2.7 | 0.0 | 52K | 2 | 30.0 hr | 2.7× |
| + Pruning + Sampling (10K steps) | 8.5 | 4.7 | 29.4 | 2.4 | 2.1 | 10K | 2 | 6.4 hr | 12.5× |
| + Pruning + LLM-Judge (10K steps) | 6.7 | 5.2 | 29.3 | **4.5** | 2.1 | 10K | 2 | 6.4 hr | 12.5× |
| **+ WEASEL (10K steps)** | **11.5** | **5.5** | **30.6** | **4.5** | **3.0** | 10K | 2 | 6.4 hr | 12.5× |
| *Qwen3-8B* | 16.4 | 18.0 | 61.1 | 35.2 | 1.7 | – | – | – | – |
| + Full (52K steps) | 17.7 | 18.2 | 59.4 | 33.3 | 2.1 | 52K | 2 | 88.5 hr | 1.0× |
| + Pruning (52K steps) | 17.6 | 12.7 | 40.3 | 15.5 | 2.6 | 52K | 2 | 44.0 hr | 2.0× |
| + Pruning + Sampling (10K steps) | 16.5 | 17.5 | 61.4 | 33.9 | 3.4 | 10K | 2 | 7.0 hr | 12.6× |
| + Pruning + LLM-Judge (10K steps) | 19.4 | 16.6 | **61.9** | 35.2 | 3.8 | 10K | 2 | 7.0 hr | 12.6× |
| **+ WEASEL (10K steps)** | **21.2** | **19.2** | **61.9** | **38.8** | **4.3** | 10K | 2 | 8.3 hr | 10.7× |

environment configuration and evaluation settings provided by the AgentLab framework (de Chezelles et al., 2025; Drouin et al., 2024a). We report aggregate SR over the full evaluation set for each benchmark. For WorkArena, we additionally report SR on Level-1 (L1) and Level-2 (L2) tasks to show the performance on simpler and more compositional workflows.

## 3.2. Results

Table 1 reports zero-shot transfer success rates; the agents are trained on AgentTrek and evaluated on WebArena-Lite, WebArena, MiniWob, and WorkArena (L1/L2) with no additional fine-tuning. WEASEL delivers the strongest accuracy-efficiency trade-off; it achieves the best SR across benchmarks while reducing training cost by an order of magnitude compared to full-data SFT, about 10.7-12.5× speed-up, depending on the model. The one-time preprocessing cost of WEASEL, which is amortized across training runs, is reported in Section E. Notably, WEASEL yields strong transfer performance on the harder benchmarks, e.g., reaching 19.2 SR on WebArena and 4.3 (L2) on WorkArena with Qwen3-8B, and 14.5 SR on WebArena-Lite with Qwen2.5-7B-Instruct. Under the same 10K-step training budget, WEASEL also outperforms standard baselines, including uniform subsampling and LLM-as-judge selection. Additional simple filtering baselines, including length-based heuristics, are presented in Section D. We note that for Qwen3-8B, the observed training speed-up is slightly lower than that of the other 10K baselines. This is due to the reasoning synthesis in §2.5, which uses self-generated rationales that are longer on average than the original AgentTrek traces, increasing

the token budget, but yielding performance that outperforms full fine-tuning.

Table 2 reports the results obtained by replacing the Agent-Trek training set with NNetNav-Live (Murty et al., 2025), which is constructed from real-world browser interactions. We fine-tune Qwen2.5-7B-Instruct for 4 epochs under the identical training setting and compare against full-data SFT and random subsampling. As in Table 1, WEASEL outperforms all baselines across all benchmarks. Moreover, training on 10K datapoints yields a 9.7× speed-up relative to the full 52K setting. This shows WEASEL's robustness with the changed training set.

**Pruning.** Table 3 compares pruning strategies under a fixed pruning ratio on a 10K random subset of AgentTrek. We report WebArena-Lite success rate (SR) and training speed-up. We fix the token budget across methods (32% of the original tokens). We test three pruning baselines: (1) *Prune-by-Bid*, which keeps a prefix up to the gold bid for each web state; (2) *Semantic* (Tan et al., 2025), which ranks nodes by semantic relevance to the gold action using BERTScore; and (3) *Target-centered + Semantic*, an ablation that forms the pruned state by taking the union of nodes selected by Target-centered (ours) and Semantic. Since these methods rely on node-grounded actions, we apply the same fixed-length prefix truncation to non-node actions (i.e., the first $L$ nodes). Our Target-centered method in §2.4 achieves the best SR, outperforming all baselines. Notably, all pruning variants yield the same 2× training speed-up relative to the unpruned data (i.e., cutting training time by 50%). Please refer to §G for more details.

*Table 2.* Success rates (SR) under a different zero-shot transfer setting where NNetNav-Live dataset (Murty et al., 2025) is used for training instead of AgentTrek in Table 1.

| Model | WebArena-Lite SR | WebArena SR | MiniWob SR | WorkArena L1 SR | WorkArena L2 SR | Training Speed-up |
|---|---|---|---|---|---|---|
| *Qwen2.5-7B-Instruct* | 5.5 | 5.2 | **41.8** | 4.8 | 0.0 | – |
| + Full (52K steps) | 10.9 | 6.9 | 38.9 | 5.2 | 6.4 | 1.0× |
| + Pruning (52K steps) | 5.5 | 3.2 | 38.4 | 1.8 | 1.3 | 1.7× |
| + Pruning + Sampling (10K steps) | 10.3 | 7.4 | 32.3 | 7.0 | 6.0 | 9.7× |
| **+ WEASEL (10K steps)** | **12.1** | **8.3** | **41.8** | **7.6** | 6.8 | 9.7× |

*Table 3.* Comparison of state-truncation strategies under the same budget (about 32% of the original tokens). We fine-tune Qwen2.5-7B-Instruct on sampled 10K datapoints of AgentTrek, and report success rates (SR) on WebArena-Lite. Training speed-up is relative to the original (untruncated) dataset.

| Method | WebArena-Lite | Speed-up |
|---|---|---|
| Original Data | 10.3 | 1.0× |
| Prune-by-Bid | 8.5 | 2.0× |
| Semantic (Tan et al., 2025) | 9.1 | 2.0× |
| Target-centered + Semantic | 7.3 | 2.0× |
| Target-centered (Ours) | **10.9** | 2.0× |

*Table 4.* Ablation on isolating the effects of data selection and reasoning synthesis (RS). All fine-tuned Qwen3-8B variants are trained on 10K data points for 2 epochs and evaluated on WebArena-Lite.

| Method | Data Selection | Reasoning Synthesis | WebArena -Lite |
|---|---|---|---|
| *Qwen3-8B* | – | – | 16.4 |
| SFT (Random) | ✗ | ✗ | 16.5 |
| SFT (Random + RS) | ✗ | ✓ | 18.2 |
| WEASEL (w/o RS ) | ✓ | ✗ | 17.0 |
| WEASEL (Ours) | ✓ | ✓ | **21.2** |

**Reasoning Synthesis.** Table 4 reports an ablation on Qwen3-8B on WebArena-Lite, isolating the effects of data selection and reasoning synthesis (RS) in §2.5. For a fair comparison, all variants are fine-tuned under the same training setting with 2 epochs on a 10K subset of AgentTrek. Our experiments show that combining data selection with RS. WEASEL achieves the best result, reaching 21.2% SR, indicating that the two components are complementary.

### 3.3. Analyses

**Importance Score.** Table 5 ablates the unary importance score $\Phi$ on WebArena-Lite (Liu et al., 2024; Zhou et al., 2024a) by varying the text paired with the goal when computing $\Phi$. For a fair comparison, all variants fine-tune Qwen2.5-7B-Instruct for 4 epochs under the same training setting. Using the goal-state pairing (ours) yields the best transfer performance, 14.5 SR, outperforming goal-reasoning, and goal-state summary, where state summaries

*Table 5.* Ablation of the unary importance score $\Phi$ on WebArena-Lite. We compare different text paired with the goal when computing $\Phi$, including the state, the reasoning trace, and a summary of the state.

| Importance term | WebArena-Lite |
|---|---|
| *Qwen2.5-7B-Instruct* | 5.5 |
| Goal-State summary | 12.7 |
| Goal-Reasoning | 9.1 |
| Goal-State (Ours) | **14.5** |

*Table 6.* Ablation of the pairwise diversity term $D(i,j)$ on WebArena-Lite. We compare using diversity computed from (1) states only ($\delta(s_i, s_j)$), (2) model answer only ($\delta(y_i, y_j)$), and (3) their max-composition $D(i,j) = \max(\delta(s_i, s_j), \delta(y_i, y_j))$.

| Diversity term | WebArena-Lite |
|---|---|
| *Qwen2.5-7B-Instrcut* | 5.5 |
| State-only | 9.7 |
| Reasoning-only | 13.9 |
| WEASEL (Ours) | **14.5** |

are generated with Flan-T5-XL (Chung et al., 2022).

**Diversity Score.** In Table 6, we ablate the pairwise diversity term by isolating state diversity, model answer diversity, and their max-composition. All results are obtaiend by fine-tuning Qwen2.5-7B-Instruct for 4 epochs, changing only the definition of $D(i,j)$. State-only diversity yields 9.7 SR on WebArena-Lite, model answer-only improves to 13.9, and the max-composition performs the best at 14.5. This indicates the two signals are complementary; state diversity promotes coverage over UI and interaction contexts, while model answer diversity captures diverse intents and rationales; taking the maximum preserves diversity in either space, reducing redundancy and improving generalization.

**Effect of Unary and Binary term.** Table 7 examines our data selection objective by isolating the unary importance term $\Phi$ and the pairwise diversity term $D$. All variants fine-tune Qwen2.5-Instruct-7B under the same training setting for 4 epochs, with the only change of objective terms. Selecting data using the unary term alone improves performance to 10.9, while using the diversity term alone yields a smaller gain of 7.9. Combining both terms in WEASEL achieves

*Table 7.* Ablation of the data selection objective on WebArena-Lite. We isolate the unary importance term ($\Phi$) and the pairwise diversity term ($D$) in our subset selection objective.

| Method | Unary $\Phi$ | Pairwise $D$ | WebArena-Lite |
|---|---|---|---|
| Qwen2.5-Instrcut-7B | – | – | 5.5 |
| SFT (+ Unary) | ✓ | ✗ | 10.9 |
| SFT (+ Diversity) | ✗ | ✓ | 7.9 |
| WEASEL (Ours) | ✓ | ✓ | **14.5** |

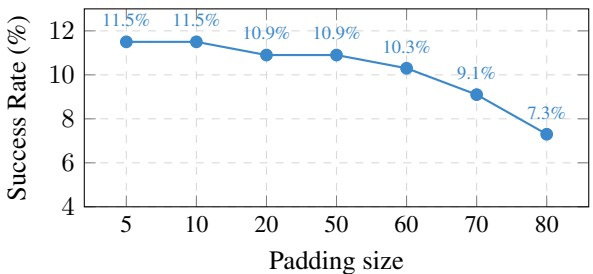

*Figure 5.* Success rate decreases as the pruning offset increases. Results are reported on WebArena-Lite.

the best result at 14.5 SR, indicating that importance and diversity provide complementary signals for constructing an effective training subset.

**Greedy Approximation Quality.** As discussed in §2.3, the 2-approximation guarantee for max-sum diversification with metric distances does not directly apply to our objective, since our diversity term $D(i, j)$ is based on semantic pseudo-distances and max-composition, and therefore does not satisfy metric properties. We therefore empirically evaluate how close the greedy solution is to the global optimum on NNetNav. For each trajectory with length between 10 and 37, we enumerate all $\binom{T}{T_0}$ subsets with $T_0 = 3$ and compute the exact objective value in Eq. (1). Across 1,877 trajectories, this yields up to $\binom{37}{3} = 7{,}770$ candidate subsets per trajectory. The greedy solution matches the exact optimum in more than 96% of trajectories and falls within the top 1% of all candidate subsets in 99.7% of trajectories. The ratio between the greedy objective value and the optimal objective value is $0.9999 \pm 0.0005$. Since the algorithm is deterministic given fixed scores and tie-breaking, the selected subsets are also stable across repeated runs. These results show that, despite the lack of a formal metric approximation guarantee, greedy selection is near-optimal in practice for our importance-diversity objective.

**Effect of Target-centered Pruning.** In Figure 5, we further validate the effect of locality in our target-centered pruning strategy (§2.4). We shift the retained context away from the target index $k_t^*$ while keeping the token budget identical. Concretely, for an offset $o \geq 0$, we define

$$\mathcal{K}_t^{(o)} = \Big( \big[\, k_t^* - o - w,\; k_t^* - o \,\big] \cup \{k_t^*\} \cup$$
$$\big[\, k_t^* + o,\; k_t^* + o + w \,\big] \Big) \cap \{1, \dots, K_t\}, \quad (10)$$

and keep $\tilde{V}_t^{(o)} = [\, v_{t,k} : k \in \mathcal{K}_t^{(o)} \,]$, and report the success rate in Figure 5. As $o$ increases, the success rate decreases roughly linearly, with random pruning performing the worst. This supports that AXTree elements near the target action are the most informative, and it motivates our design of target-centered pruning.

*Table 8.* Transfer of WEASEL to a multimodal GUI-agent setting on Android in the Wild (AITW). We evaluate on a held-out 500-example test subset.

| Method | AITW Acc. (%) |
|---|---|
| Qwen2.5-VL-3B-Instruct | 4.4 |
| Random selection | 5.8 |
| WEASEL greedy selection | **6.6** |

**Transfer beyond AXTree-based Web Agents.** Although our main experiments focus on text-based web agents with linearized AXTree observations, we further evaluate whether the selection component of WEASEL transfers to a multimodal GUI-agent setting. We conduct an additional experiment on Android in the Wild (AITW) (Rawles et al., 2023), where agents observe screenshots and execute mobile UI actions, without AXTree linearization or target-centered AXTree pruning. We keep the greedy trajectory-selection algorithm unchanged and replace only the scoring modules with multimodal ones: for diversity, we compute state similarity using SigLIP2-Large-384 (Tschannen et al., 2025) screenshot embeddings, and for importance, we compute goal-text/screenshot alignment using CLIP ViT-L/14-336 (Radford et al., 2021). Starting from a 10K training pool constructed from the official AITW training split, we compare matched-size 3.1K subsets selected by uniform random sampling and WEASEL. Using Qwen2.5-VL-3B-Instruct as the base policy model, WEASEL improves accuracy on a held-out 500-example AITW test subset from 5.8% with random selection to 6.6%, compared to 4.4% for the base model, as shown in Table 8. These preliminary results suggest that WEASEL is not tied to AXTree-based web states or text-only inputs, although further exploration of multimodal scoring choices may yield stronger configurations for GUI agents.

We further analyze the sensitivity of WEASEL to key hyperparameters, including the importance-diversity trade-off $\lambda$, pruning window size $w$, and selection budget $T_0$, in Section C.

## 4. Related Work

**Training Data and Environment for Web Agents.** Early progress on LLM-based web agents has been driven by training with online interaction and reinforcement learning. WebRL (Qi et al., 2024; 2025), TTI (Shen et al., 2025), WebAgent-R1 (Wei et al., 2025), and AutoWebGLM (Lai et al., 2024) improve agent performance through interactive training and environment rollouts. However, these approaches are often developed and evaluated in in-benchmark settings, which may not fully reflect performance under distribution shift.

More recently, large-scale foundation datasets have enabled scaling supervision for web agent training. AgentTrek (Xu et al., 2025b) synthesizes large-scale web agent trajectories from tutorial resources, while NNetNav (Murty et al., 2025) collects broad supervision from real-world browsing and retroactive labeling. Despite this progress, agents trained on such data can still exhibit substantial performance drops in out-of-domain test settings when evaluated on websites and interaction patterns that differ from those seen during training (Murty et al., 2025). Our work complements these efforts by treating out-of-domain transfer and training efficiency as first-class goals of *data selection*, selecting a fixed-budget subset that is both goal-relevant and diverse to reduce redundancy and improve robustness under distribution shift.

**Data Selection and State Pruning for Web Agents.** Selecting an informative subset of training data is a classic problem in machine learning, with coreset and data subset selection methods studied for efficiency and robustness (Sener & Savarese, 2018; Mirzasoleiman et al., 2020; Killamsetty et al., 2021). Related ideas have also been explored for preference optimization of LLMs, where careful curation can reduce training cost while improving alignment (Deng et al., 2025). Similar trends appear in instruction tuning and pretraining, where curated subsets or optimized data mixtures can match or improve performance at lower cost (Xie et al., 2023; Bukharin et al., 2024).

To our knowledge, we propose WEASEL, the first principled subset selection method tailored to Web agent training, which improves out-of-domain generalization and training efficiency by jointly prioritizing goal-relevant steps and promoting diversity among selected steps. Several lines of work reduce web observation states primarily for test-time efficiency or context limitations, for example via learned contextualization/summarization modules or LLM-based retrieval over AXTree lines (Lee et al., 2025; Kerboua et al., 2025a;b). Our target-centered pruning instead targets training efficiency and trims states without introducing any additional modules or parameters.

## 5. Conclusion

We presented WEASEL, a data selection framework for compute-efficient offline web agent training that explicitly targets out-of-domain transfer. WEASEL selects a fixed-budget subset of trajectory steps by optimizing a simple objective that balances goal-conditioned importance and pairwise diversity, solved efficiently with a greedy algorithm. In addition, we introduced target-centered AXTree pruning to further reduce redundant input context, and self-reasoning synthesis to better fine-tune reasoning-native models. Across multiple web agent benchmarks and model families, WEASEL consistently improves zero-shot transfer performance under domain shift while substantially reducing training cost. Notably, training with only 19% of the original training data achieves comparable or better performance than full-data SFT, while delivering roughly 9.7-12.5× speed-ups, making offline Web agent training more practical at scale. More broadly, our results suggest that careful step-level curation and lightweight observation pruning can be complementary to scaling data, enabling stronger generalization without requiring additional modules or complex test-time components.

## Impact Statement

Our work makes offline training of web agents more practical by reducing the amount of redundant supervision needed to achieve strong transfer to unseen websites and interaction patterns. By selecting a small but informative subset of trajectory steps and pruning irrelevant state context, WEASEL can substantially lower compute and energy costs for training and iteration, which may help broaden access to web agent research beyond groups with large training budgets.

At the same time, improving web agent capability can increase the risk of misuse (e.g., automating spam, scraping, or unauthorized actions on websites). We encourage responsible release practices, including clear usage policies, rate limits, and safety evaluations on harmful or policy violating tasks, and we recommend deploying agents with safeguards such as action constraints and human oversight for high stakes.

## Acknowledgements

This work was financially supported by Institute of Information & Communications Technology Planning & Evaluation (IITP) grant funded by the Korea government (MSIT) (No. RS-2019-II191082, SW StarLab, No. RS-2022-II220156, Fundamental research on continual meta-learning for quality enhancement of casual videos and their 3D metaverse transformation, No. RS-2025-25442338, AI star Fellowship Support Program(Seoul National Univ.) and No. RS-2021-II211343, Artificial Intelligence Grad-

uate School Program (Seoul National University)) and the InnoCORE program of the Ministry of Science and ICT (N10250156). Xing Han Lù acknowledges the support of the Natural Sciences and Engineering Research Council of Canada (NSERC) [funding reference no. 579403]. Gunhee Kim is the corresponding author.

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

## A. Experimental Details

**Training setup and hyperparameters** We report the hyperparameters for training WEASEL in Table 9. We fine-tune base models with LoRA (Hu et al., 2021) adapter with the configuration in Table 9. For computing BERTScore similarities used in WEASEL, we use roberta-large (Liu et al., 2019) as the encoder. After applying WEASEL with $T_0 = 3$ to each dataset (AgentTrek and NNetNav-Live), we obtain 12K selected datapoints from AgentTrek (Xu et al., 2024) and 16K from NNetNav-Live (Murty et al., 2025); for consistency across datasets, we uniformly sample 10K datapoints from each for training.

*Table 9.* Training hyperparameters for fine-tuning Qwen2.5-7B-Instruct, Gemma3-4B-IT, and Qwen3-8B on AgentTrek.

| Hyperparameter | Qwen2.5-7B-Instruct | Gemma3-4B-IT | Qwen3-8B |
|---|---|---|---|
| Training epochs | 4 | 2 | 2 |
| Training set size | 10K | 10K | 10K |
| Batch size | 8 | 16 | 8 |
| Optimizer | AdamW | AdamW | AdamW |
| Learning rate | 2e-5 | 2e-5 | 1e-6 |
| LR schedule | Cosine | Cosine | Cosine |
| Mixed precision | BF16 | BF16 | BF16 |
| LoRA rank | 8 | 8 | 8 |
| LoRA alpha | 8 | 8 | 8 |
| LoRA dropout | 0.0 | 0.0 | 0.0 |

## B. Alternative Similarity Model for $\Phi$ and $D$

Table 10 evaluates how the choice of similarity encoder affects WEASEL by replacing BERTScore with a learned embedding model for computing both $\Phi$ and $D$. Under the same 10K-step training budget and identical hyperparameters, the embedding-based variant Qwen3-Embedding-0.6B and Qwen3-Embedding-4B (Zhang et al., 2025) still substantially outperforms the SFT baseline with random subsampling, indicating that WEASEL is not sensitive to the similarity implementation, and the gains are robust. In our experiments, BERTScore with roberta-large (Liu et al., 2019) is the default in the main results.

*Table 10.* Effect of the similarity model used in WEASEL for computing $\Phi$ and $D$ on WebArena-Lite. All 10K runs use the same training hyperparameters; only the similarity encoder is changed.

| Method | # Steps | WebArena-Lite SR |
|---|---|---|
| *Qwen2.5-7B-Instrcut* | – | 5.5 |
| SFT (Random) | 10K | 9.1 |
| WEASEL (Embedding; `Qwen3-Embedding-0.6B`) | 10K | 12.7 |
| WEASEL (Embedding; `Qwen3-Embedding-4B`) | 10K | 14.5 |
| WEASEL (BERTScore; `roberta-large`) | 10K | 14.5 |

## C. Hyperparameter Sensitivity

We analyze the sensitivity of WEASEL to key design hyperparameters, including the importance-diversity trade-off $\lambda$, the pruning window size $w$, and the selection budget $T_0$. In the main experiments, we use $\lambda = 1$, $w = 60$, and $T_0 = 3$. We additionally evaluate alternative settings by varying $\lambda$, changing the pruning window size, and selecting different numbers of steps per trajectory, where $T$ denotes the trajectory length.

As shown in Table 11, WEASEL consistently outperforms the standard baselines across a broad range of hyperparameter choices. Although the default setting achieves the best overall performance, the gains remain robust under different trade-off weights, pruning windows, and selection budgets. These results indicate that WEASEL is not overly sensitive to these design choices.

## D. Additional Filtering Baselines

We further compare WEASEL with simple filtering heuristics that do not use our importance-diversity objective. In addition to uniform random subsampling and LLM-as-judge filtering, we evaluate two length-based filtering baselines, motivated

*Table 11.* Sensitivity of WEASEL to key hyperparameters on WebArena-Lite. We vary the importance-diversity trade-off $\lambda$, pruning window size $w$, and selection budget $T_0$.

| Method | WebArena-Lite SR |
|---|---|
| Qwen2.5-7B-Instruct | 5.5 |
| SFT (Random) | 9.1 |
| SFT (LLM-as-Judge) | 8.5 |
| WEASEL ($\lambda = 0.5$) | 10.9 |
| WEASEL ($\lambda = 2.0$) | 12.1 |
| WEASEL ($\lambda = 4.0$) | 10.3 |
| WEASEL ($w = 40$) | 12.7 |
| WEASEL ($w = 80$) | 12.8 |
| WEASEL ($T_0 = 0.25T$) | 12.1 |
| WEASEL (default) | **14.5** |

*Table 12.* Comparison with simple filtering baselines on WebArena-Lite. All SFT variants use 10K selected datapoints from AgentTrek.

| Method | WebArena-Lite |
|---|---|
| Qwen2.5-7B-Instruct | 5.5 |
| SFT (Random, 10K) | 9.1 |
| SFT (LLM-as-Judge, 10K) | 8.5 |
| SFT (Length-sorted indices 5K–15K) | 5.5 |
| SFT (Length-sorted indices 35K–45K) | 10.9 |
| WEASEL (10K) | **14.5** |

by curriculum-style selection where trajectory length is used as a proxy for difficulty. Specifically, we sort the AgentTrek dataset by trajectory length and retain datapoints whose sorted indices fall within a fixed range. We consider indices 5K–15K and 35K–45K, yielding 10K datapoints in each case.

As shown in Table 12, length-based filtering improves over some simple baselines when selecting later, longer trajectories, but still underperforms WEASEL. This suggests that trajectory length alone is not sufficient for selecting transferable supervision; balancing goal relevance with diversity provides a stronger signal for out-of-domain generalization.

## E. Preprocessing Cost

WEASEL incurs a one-time preprocessing cost before training, after which the selected subset can be reused across training runs without additional selection overhead. On the full AgentTrek dataset, the preprocessing pipeline takes 7.7 hours using RTX 6000 GPUs, including importance computation, diversity computation, greedy subset selection, and reasoning synthesis. This cost is substantially smaller than full-data SFT, which takes 136 hours on 4 RTX 6000 GPUs, while enabling the training speedups reported in Table 1. As shown in Table 13, the main preprocessing cost comes from reasoning synthesis, whereas importance-diversity scoring and greedy selection are relatively lightweight.

## F. Prompt Used for Experiments

### F.1. LLM-as-a-Judge Experiment Details

We use the following prompt to score each datapoint for the LLM-as-a-judge filtering step. For generation, we set the sampling parameters to temperature of 0.0 and top-$p$ of 0.9. After generating a score, we group datapoints by their assigned score and construct the final set by sampling equally from scores 4 and 5, yielding 10K datapoints in total.

*Table 13.* Cost breakdown of the WEASEL preprocessing pipeline on the full AgentTrek dataset. Preprocessing is a one-time cost; the selected subset can be reused for multiple training runs.

| Stage | Wall-clock Time | GPU Type | # GPUs |
|---|---|---|---|
| Importance computation | 0.35 h | RTX 6000 | 1 |
| Diversity computation | 0.50 h | RTX 6000 | 1 |
| Greedy algorithm | < 0.02 h | – | – |
| Reasoning synthesis | 7.5 h | RTX 6000 | 4 |

**Prompt for LLM-as-a-Judge**

```
You are evaluating how useful a single state-action step is for solving a web task.

Task goal:
{GOAL}

State + action description (AXTree + history + assistant output):
{STATE_BLOCK}

On a scale from 1 to 5, where
- 1 = not helpful at all for achieving the goal,
- 2 = slightly helpful but not critical,
- 3 = somewhat helpful but not critical,
- 4 = helpful and critical,
- 5 = very helpful or directly crucial,
rate how much this step helps to solve the goal.

Provide 2-3 concise sentences of reasoning that cite specific observations/actions.
End with a line exactly like:
Score: <number between 1 and 5>
```

## F.2. Self-Reasoning Synthesis Details

We use the prompt shown in Appendix X to self-synthesize reasoning. The prompt is largely adapted from the BrowserGym-style instructions (de Chezelles et al., 2025). For generation, we set the sampling parameters to temperature of 0.2 and top-$p$ of 0.9. After generation, we parse the model output and replace the original datasets `think` and `memory` blocks with those extracted from the generated response.

**Prompt for reasoning synthesis**

```
# General Instructions

You are a UI Assistant, your goal is to help the user perform tasks using a web
browser. You can communicate with the user via a chat, in which the user gives you
instructions and in which you can send back messages. You have access to a web
browser that both you and the user can see, and with which only you can interact via
specific commands.

Review the instructions from the user, the current state of the page and all other
information to find the best possible next action to accomplish your goal. Your
answer will be interpreted and executed by a program, make sure to follow the
formatting instructions.

## Goal:
{GOAL}

# Observation of current step:
```

```
## AXTree: {STATE_BLOCK}
# History of interaction with the task:
{HISTORY}

# Action space:

16 different types of actions are available.

noop(wait_ms: float = 1000)
    Description: Do nothing, and optionally wait for the given time (in
    milliseconds).
    Examples:
        noop()

        noop(500)

send_msg_to_user(text: str)
    Description: Sends a message to the user.
    Examples:
        send_msg_to_user('Based on the results of my search, the city was built in
        1751.')

scroll(delta_x: float, delta_y: float)
    Description: Scroll horizontally and vertically. Amounts in pixels, positive for
    right or down scrolling, negative for left or up scrolling. Dispatches a wheel
    event.
    Examples:
        scroll(0, 200)

        scroll(-50.2, -100.5)

fill(bid: str, value: str)
    Description: Fill out a form field. It focuses the element and triggers an input
    event with the entered text. It works for <input>, <textarea> and
    [contenteditable] elements.
    Examples:
        fill('237', 'example value')

        fill('45', 'multi-line\nexample')

        fill('a12', 'example with "quotes"')

select_option(bid: str, options: str | list[str])
    Description: Select one or multiple options in a <select> element. You can
    specify option value or label to select. Multiple options can be selected.
    Examples:
        select_option('a48', 'blue')

        select_option('c48', ['red', 'green', 'blue'])

click(bid: str, button: Literal['left', 'middle', 'right'] = 'left', modifiers:
list[typing.Literal['Alt', 'Control', 'Meta', 'Shift']] = [])
    Description: Click an element.
    Examples:
        click('a51')

        click('b22', button='right')

        click('48', button='middle', modifiers=['Shift'])

dblclick(bid: str, button: Literal['left', 'middle', 'right'] = 'left', modifiers:
list[typing.Literal['Alt', 'Control', 'Meta', 'Shift']] = [])
    Description: Double click an element.
```

```
    Examples:
        dblclick('12')

        dblclick('ca42', button='right')

        dblclick('178', button='middle', modifiers=['Shift'])

hover(bid: str)
    Description: Hover over an element.
    Examples:
        hover('b8')

press(bid: str, key_comb: str)
    Description: Focus the matching element and press a combination of keys. It
    accepts the logical key names that are emitted in the keyboardEvent.key property
    of the keyboard events: Backquote, Minus, Equal, Backslash, Backspace, Tab,
    Delete, Escape, ArrowDown, End, Enter, Home, Insert, PageDown, PageUp,
    ArrowRight, ArrowUp, F1 - F12, Digit0 - Digit9, KeyA - KeyZ, etc. You can
    alternatively specify a single character you'd like to produce such as "a" or
    "#". Following modification shortcuts are also supported: Shift, Control, Alt,
    Meta.
    Examples:
        press('88', 'Backspace')

        press('a26', 'Control+a')

        press('a61', 'Meta+Shift+t')

focus(bid: str)
    Description: Focus the matching element.
    Examples:
        focus('b455')

clear(bid: str)
    Description: Clear the input field.
    Examples:
        clear('996')

drag_and_drop(from_bid: str, to_bid: str)
    Description: Perform a drag & drop. Hover the element that will be dragged.
    Press left mouse button. Move mouse to the element that will receive the drop.
    Release left mouse button.
    Examples:
        drag_and_drop('56', '498')

upload_file(bid: str, file: str | list[str])
    Description: Click an element and wait for a "filechooser" event, then select
    one or multiple input files for upload. Relative file paths are resolved
    relative to the current working directory. An empty list clears the selected
    files.
    Examples:
        upload_file('572', 'my_receipt.pdf')

        upload_file('63', ['/home/bob/Documents/image.jpg',
        '/home/bob/Documents/file.zip'])

go_back()
    Description: Navigate to the previous page in history.
    Examples:
        go_back()

go_forward()
    Description: Navigate to the next page in history.
    Examples:
```

```
        go_forward()

goto(url: str)
    Description: Navigate to a url.
    Examples:
        goto('http://www.example.com')

Only a single action can be provided at once. Example:
fill('a12', 'example with "quotes"')

# Abstract Example

Here is an abstract version of the answer with description of the content of
each tag. Make sure you follow this structure, but replace the content with your
answer:

<think>
Think step by step. If you need to make calculations such as coordinates, write them
here. Describe the effect
that your previous action had on the current content of the page.
</think>

<memory>
Write down anything you need to remember for next steps. You will be presented
with the list of previous memories and past actions.
</memory>

<action>
One single action to be executed. You can only use one action at a time.
</action>

# Concrete Example

Here is a concrete example of how to format your answer.
Make sure to follow the template with proper tags:

<think>
My memory says that I filled the first name and last name, but I can't see any
content in the form. I need to explore different ways to fill the form. Perhaps
the form is not visible yet or some fields are disabled. I need to replan.
</think>

<memory>
I clicked on bid 32 to activate tab 2. The accessibility tree should mention
focusable for elements of the form at next step.
</memory>

<action>
fill('a12', 'example with "quotes"')
</action>

# Instructions and Guidelines
You must regenerate the assistant's reasoning (<think>) and short memory
(<memory>) for the next step.

- The <think> block is your internal reasoning about what to do next.
- The <memory> block is a very short summary (12 sentences) of important information
  that should be carried over to the next step.
- The <memory> block MUST NOT be empty. Always write at least one sentence.
- Do NOT copy the whole conversation into <memory>. Only keep what is useful later.
- Use the action provided below verbatim inside <action>. Do not change or omit it.
- The action is the next step to execute; assume it has NOT been done yet.
  Reason about why this action should be taken next, rather than describing
```

```
   it as already completed.

Action to execute (for your <action> block):
{action_block}

Now, for the CURRENT input, respond in EXACTLY the following format, with
no extra text before or after the tags:

<think>
[concise reasoning consistent with the context above]
</think>

<memory>
[short memory to carry over; at least one sentence]
</memory>

<action>
{action_block}
</action>
```

## G. Pruning experiment details

**Target-centered pruning.** We use a fixed window size w=60 for target-centered pruning, and a larger window of w=120 for non-node actions. For `Static` nodes that do not have an index, we exclude them from the window-size count.

**Semantic pruning.** We rank leaf nodes by semantic similarity to a query formed by concatenating the `goal` and the `gold answer`. Similarity is based on the BERTScore (Zhang et al., 2020) using the `paraphrase-mpnet-base-v2` encoder, and we retain the top 80 leaf nodes. Each node is represented by concatenating its own text with its ancestor context (i.e., the parent chain).

**Target-centered + Semantic pruning.** We first select 20 leaf nodes via semantic pruning and, independently, select 60 nodes via target-centered pruning with a window size of 30. The final node set is taken as the union of the two selected sets.

Across pruning methods, we tune the pruning hyperparameters to keep the dataset-level average token count comparable.

