# OpenReview forum: "Weasel: Out-of-Domain Generalization for Web Agents via Importance-Diversity Data Selection"
_ICML.cc/2026/Conference — ICML 2026 regular_

### Official Review · Reviewer_BwFe · 2026-03-09

**Soundness:** 3
**Presentation:** 3
**Significance:** 2
**Originality:** 3
**Overall Recommendation:** 5
**Confidence:** 4

**Summary:**

This paper is motivated by the ood generalization for web agents training under domain shift. It proposes WEASEL, a fixed-budget step-level data selection method that balances goal-conditioned importance and diversity, with AXTree pruning and reasoning synthesis. Through experiments, WEASEL is proved to be a promising and efficient web-agent training method.

**Compliance With Llm Reviewing Policy:**

Affirmed.

**Key Questions For Authors:**

1. Could the authors provide the missing ablation of WEASEL without self-reasoning synthesis, especially for Qwen3-8B?
2. How sensitive is WEASEL to key design hyperparameters, such as the importance-diversity trade-off and the pruning window size?

**Limitations:**

No. The paper discusses potential societal risks and mitigation strategies, but it does not sufficiently discuss the method’s own technical limitations.

**Strengths And Weaknesses:**

**Strengths**:
- Clear motivation. The paper is well motivated by the challenge of ood generalization in web agents with substantial redundancy in offline trajectories. The overall content is presented clearly with a coherent narrative linking step-level selection, pruning and reasoning synthesis.
- Sound experiment. The experimental section is relatively strong with multiple benchmarks, metrics and model families. And more ablation experiments strongly support the effectiveness of the method.

**Weaknesses**:
- Limited analysis. Some parts of the method would benefit from deeper analysis and clearer experimental justification, such as sensitivity to the selection budget or key hyperparameters. Also WEASEL without reasoning synthesis in Table 4 should be included to more fully prove the point.
- Confined scenario. The method is closely tied to the paper’s text-based web-agent setting, in particular the linearized AXTree state representation and access to gold action targets for pruning. As a result, it is less clear how directly the approach would transfer to other agent settings, such as GUI agents with visual inputs.

---

> ### Author Rebuttal · Authors · 2026-03-31
>
> Thank you for the positive feedback. We address your questions and concerns below.
>
> ---
>
> **Q1. Could the authors provide the missing ablation of WEASEL without self-reasoning synthesis, especially for Qwen3-8B? (W1, Q1)**
>
> We thank the reviewer for this suggestion. We conduct an ablation experiment: please refer to our **response to Q1 from Reviewer 6iYw**. The results show that WEASEL remains effective even without self-reasoning synthesis in Qwen3-8B, and still outperforms all the baseline methods.
>
> ---
>
> **Q2. How sensitive is WEASEL to key design hyperparameters, such as the importance-diversity trade-off and the pruning window size? (W1, Q2)**
>
> We thank the reviewer for this question. To evaluate sensitivity to key design hyperparameters, we conduct additional experiments by varying the importance-diversity trade-off $\lambda$, the pruning window size, and the selection budget $T_0$. Specifically, while the main paper uses $\lambda=1$, $w=60$, and $T_0=3$, we additionally test $\lambda \in {0.5, 1.5, 2.0}$, the pruning window $w \in {40, 80}$, and the selection budget with $T_0 = 0.25T$, where $T$ denotes the length of each trajectory. Across all settings, WEASEL consistently outperforms all the baseline methods, indicating that WEASEL is not overly sensitive to these hyperparameters. While the original setting remains the best overall, the gains are robust across a broad range of choices.
>
>
> | Method | WebArena-Lite |
> |---|---|
> | Qwen2.5-7B-Instruct | 6.7 |
> | SFT (Random) | 8.5 |
> | LLM-Judge | 5.5 |
> | WEASEL ($\lambda = 0.5$) | 11.5 |
> | WEASEL ($\lambda = 1.5$) | 12.1 |
> | WEASEL ($\lambda = 2.0$) | 10.3 |
> | WEASEL (pruning window = 40) | 12.7 |
> | WEASEL (pruning window = 80) | 12.8 |
> | WEASEL ($T_0 = 0.25T$) | 10.9 |
> | Original WEASEL | **13.3** |
>
> *Table 1. Sensitivity of WEASEL to key hyperparameters on WebArena-Lite.*
>
> ---
>
> **Q3. The method is closely tied to the paper’s text-based web-agent setting, in particular, the linearized AXTree state representation and access to gold action targets for pruning. As a result, it is less clear how directly the approach would transfer to other agent settings, such as GUI agents with visual inputs. (W2) (Reviewer BwFe, Reviewer 7RVZ)**
>
>
> We thank the reviewers for raising this important point. Although the main focus of this work is text-based web agents, we conduct an additional experiment on Android In The Wild (AITW) [1], a multimodal GUI-agent benchmark with screenshot observations and mobile UI actions. This setting differs from our original setup in two ways: (i) the state is visual rather than text-only, and (ii) there is no AXTree linearization. We keep the WEASEL’s trajectory-selection algorithm unchanged, and replace only the scoring modules with multimodal ones: for diversity, we compute state similarity from screenshot embeddings using SigLIP2-Large-384; for importance, we compute goal-text / screenshot alignment using CLIP ViT-L/14-336. We use Qwen2.5-VL-3B-Instruct as the base policy model. Starting from the official AITW training split, we first construct a 10K training pool, then build matched-size 3.1K training examples using either uniform random sampling or WEASEL’s greedy selection. On a held-out AITW test 500 examples, the base model achieves 4.4\% accuracy, random selection improves to 5.8\%, and WEASEL further improves to 6.6\%. These results suggest that WEASEL is not tied to AXTree-based web states or text-only inputs, but can transfer more broadly to multimodal GUI-agent settings. However, given the highly limited rebuttal period, our implementation choices could be further explored to identify improved configurations for multimodal agents.
>
>
>
> | Method | AITW Acc. (%) |
> |---|---|
> | Qwen2.5-VL-3B-Instruct | 4.4 |
> | Random selection (3.1K from 10K pool) | 5.8 |
> | WEASEL greedy selection (3.1K from 10K pool) | **6.6** |
>
> *Table 2. WEASEL generalizes beyond text-based web agents to a multimodal GUI-agent setting (AITW, 500-example held-out test subset).*
>
> ---
>
> **References**
>
> *[1] Christopher Rawles, Alice Li, Daniel Rodriguez, Oriana Riva, and Timothy Lillicrap. 2023. Android in the wild: A large-scale dataset for android device control. arXiv preprint arXiv:2307.10088.*

---

> > ### Author Rebuttal · Reviewer_BwFe · 2026-04-01
> >
> > Thanks for the author's response and all my questions have been resolved. I believe the paper should be accepted, and I will raise the score to 5.

---

> > > ### Author Response · Authors · 2026-04-02
> > >
> > > Thank you for the thoughtful feedback. We are glad the rebuttal addressed your concerns, and we sincerely appreciate your updated score.

---

### Official Review · Reviewer_7RVZ · 2026-03-10

**Soundness:** 3
**Presentation:** 2
**Significance:** 3
**Originality:** 3
**Overall Recommendation:** 3
**Confidence:** 3

**Summary:**

This paper proposes WEASEL, a data selection method to improve the generalization and efficiency of LLM-based web agents trained from offline interaction trajectories. The method selects a fixed-budget subset of trajectory steps by balancing goal relevance (importance) and behavioral diversity, using a greedy optimization algorithm. It also introduces AXTree pruning to remove irrelevant webpage content and model-consistent reasoning synthesis to align reasoning style during fine-tuning. Experiments across multiple datasets and benchmarks show improved out-of-domain performance and up to 12.5× faster training compared to standard fine-tuning.

**Compliance With Llm Reviewing Policy:**

Affirmed.

**Key Questions For Authors:**

1. How sensitive is the method to the way importance and diversity are defined?
A lot seems to depend on these similarity scores. If different choices give very different results, that would make the method feel less stable.

2. Can you separate more clearly what comes from trajectory selection, AXTree pruning, and reasoning alignment?
Right now these parts are a bit mixed together. Clearer ablations would help me understand what is actually doing the work.

3. How much of the gain is from better data efficiency, and how much is from actual improvement in agent quality?
This would change how I view the contribution, since saving training cost and improving capability are not exactly the same thing.

4. How would this method perform in settings beyond the benchmarks used here?
The results look useful on web-agent tasks, but it is less clear whether the same idea would hold up in more different environments.

5. How does WEASEL compare to simpler filtering baselines?
If a much simpler selection rule works almost as well, then the practical advantage of the full method would be less convincing.

**Limitations:**

yes

**Strengths And Weaknesses:**

For Strengths:

Soundness: The method is fairly reasonable. Using importance and diversity to select useful trajectory steps makes sense, and the greedy algorithm is a practical choice.

Presentation: The paper is generally clear and not hard to follow. The main idea and motivation are explained well.

Significance: The problem is useful and relevant, especially because web-agent training data can be noisy and repetitive.

Originality: The paper is not fully novel at the component level, but the way these ideas are put together for web agents is still meaningful.

For Weaknesses:

Soundness: Some parts feel a bit heuristic, especially how importance and diversity are measured. The paper could give more analysis on whether these choices are robust.

Presentation: The three parts of the method are somewhat mixed together, so it is a little hard to see which part contributes most.

Significance: The work mainly improves data selection and efficiency, rather than the agent’s core reasoning or planning ability.

Originality: Most of the building blocks already exist in other forms, so the novelty is more in the combination than in a truly new method.

---

> ### Author Rebuttal · Authors · 2026-03-31
>
> We appreciate the detailed feedback. We address all the concerns below.
>
> ---
> **Q1. How sensitive is the method to the way importance and diversity are defined? A lot seems to depend on these similarity scores. If different choices give very different results, that would make the method feel less stable. (Reviewers 7RVZ, w9xm)**
>
> We thank the reviewer for this question. WEASEL is not tied to a specific similarity measure for importance scores $\Phi(t)$ and pairwise diversity scores $D(i,j)$. As shown in Appendix B, replacing BERTScore with Qwen3-Embedding-4B for both $\Phi(t)$ and $D(i,j)$ still yields strong performance and continues to outperform all the baselines. We further evaluate a smaller embedding model Qwen3-Embedding-0.6B to compute both $\Phi(t)$ and $D(i,j)$, which achieves 12.7 on WebArena-Lite. Across all similarity models we have tested, WEASEL consistently outperforms the baselines by significant margins. We believe that these results suggest that the method is not sensitive to the similarity implementation, and the gains are robust.
>
> | Method | WebArena-Lite |
> |---|---|
> |Qwen2.5-7B-Instruct|6.7|
> |SFT (Random)|8.5|
> |LLM-Judge|5.5|
> |WEASEL (Embedding: Qwen3-Embedding-0.6B)|12.7|
> |WEASEL (Embedding: Qwen3-Embedding-4B)|12.1|
> |WEASEL (BERTScore: roberta-large)|**13.3**|
>
> *Table 1. Effect of the similarity model used to compute $\Phi(t)$ and $D(i,j)$ on WebArena-Lite.*
>
> ---
>
> **Q2. Can you separate more clearly what comes from trajectory selection, AXTree pruning, and reasoning alignment?**
>
> Please refer to our **response to Q1 from Reviewer 6iYw**, where we present an ablation study on WebArena-Lite (Qwen3-8B) isolating the effects of data selection and reasoning synthesis. The results show that each component yields meaningful gains, and their combination achieves the best performance.
>
> ---
>
> **Q3. How much of the gain is from better data efficiency, and how much is from actual improvement in agent quality?**
>
> We would like to highlight that WEASEL provides independent gains in both data efficiency and generalization capability, via the interplay of its three components: (1) target-centered state pruning, (2) importance-diversity data selection, and (3) self-reasoning synthesis.
>
> Each component contributes differently to efficiency and performance. We curate a subset of results from the paper below, highlighting the contribution of each component. (1) Target-centered state pruning primarily improves training efficiency (2.2x speedup), while incurring a slight performance tradeoff. (2) Data selection, when combined with pruning, contributes to both aspects simultaneously, yielding a 12.6x speedup alongside a notable performance increase. (3) Self-reasoning synthesis mainly boosts generalization performance at a minor cost to the training speedup.
>
> All together, Weasel provides advantages in both aspects, achieving both a big training speedup while also surpassing full fine-tuning in task success rate.
>
> | Method | Training Speedup | WebArena-Lite |
> |---|---|---|
> | Qwen3-8B |-|13.5|
> | Full Fine-tuning |1.0x|16.4|
> | + (1) | 2.2x |15.2|
> | + (1) + (2) |12.6x|18.2|
> | + (1) + (2) + (3) (WEASEL) | 11.3x | **20.0** |
>
> *Table 2. Contribution of each WEASEL component on Qwen3-8B.*
>
> ---
>
> **Q4. How would this method perform in settings beyond the benchmarks used here? The results look useful on web-agent tasks, but it is less clear whether the same idea would hold up in more different environments.**
>
>
> We thank the reviewer for the valuable feedback. To address this concern, we conducted an additional experiment in a GUI mobile-agent domain with visual inputs; please refer to our **response to Q3 from Reviewer BwFe**. The results suggest that WEASEL is not confined to text-based web-agent settings, but can generalize more broadly as a trajectory-selection framework to datasets with different input modalities. We will include these results in the revision.
>
> ---
>
> **Q5. How does WEASEL compare to simpler filtering baselines?**
>
> We already compare WEASEL against several simple baselines, including uniform random subsampling, LLM-as-Judge filtering, and selection based only on goal-observation semantic similarity, and WEASEL consistently performs the best.
>
> We further evaluate two simple length-based filtering baselines, motivated by curriculum learning where length is often used as a proxy for difficulty. We sort the AgentTrek dataset by length and keep only datapoints within a selected index range. We keep datapoints with sorted indices 5K–15K in the first experiment, and 35K–45K in the second. Each filtered set contains 10K datapoints. In both cases, WEASEL still outperforms all these heuristics.
>
>
> | Method | WebArena-Lite |
> |---|---|
> | Qwen2.5-7B-Instruct | 6.7|
> | SFT (Random, 10K) | 8.5|
> | SFT (LLM-as-Judge, 10K) | 5.5|
> | SFT (Length-sorted indices 5K–15K) | 5.5 |
> | SFT (Length-sorted indices 35K–45K) |10.9 |
> | WEASEL (10K) |**13.3**|
>
> *Table 3. Comparison with simple filtering baselines on WebArena-Lite.*

---

> > ### Author Rebuttal · Reviewer_7RVZ · 2026-04-04
> >
> > Seems that target centered state pruning gives poor performance and efficiency, why not remove them in the final approach?

---

> > > ### Author Response · Authors · 2026-04-06
> > >
> > > **Q. Seems that target centered state pruning gives poor performance and efficiency, why not remove them in the final approach?**
> > >
> > > We thank you for the continued feedback and engagement. We would like to respectfully clarify that target-centered pruning offers a meaningful training efficiency gain with minimal performance impact. For instance, on Qwen2.5-7B-Instruct, it reduces training time from 136 hr to 62 hr (2.2x speedup) with no degradation in WebArena-Lite success rate. On Gemma3-4B-IT, it reduces training time from 80 hr to 30 hr (2.7x speedup) with only a 0.6%p accuracy drop (Table 1 in the paper).
> > >
> > > Importantly, target-centered pruning remains effective when combined with the other components of our method. To demonstrate this, we ran an additional ablation on Qwen3-8B, removing target-centered pruning from WEASEL while retaining greedy data selection and self-reasoning synthesis. The results show that adding target-centered pruning improves efficiency (2.14x speedup) while retaining performance (+0.6%p), confirming that pruning web states to their most relevant parts benefits training on higher-quality data subsets as well.
> > >
> > > | Method | Training Speedup | WebArena-Lite |
> > > |---|---|---|
> > > | Qwen3-8B | - | 13.5 |
> > > | Full Fine-tuning | 1.0x | 16.4 |
> > > | WEASEL w/o target-centered pruning | 5.0x | 19.4 |
> > > | WEASEL| 10.7x | **20.0** |
> > >
> > > *Table 3. Ablation of target-centered pruning from WEASEL.*

---

### Official Review · Reviewer_6iYw · 2026-03-13

**Soundness:** 3
**Presentation:** 3
**Significance:** 3
**Originality:** 3
**Overall Recommendation:** 5
**Confidence:** 3

**Summary:**

In this paper, the authors proposed Weasel, a trajectory selection method of web agents. The pipeline selects subsets of trajectory steps by optimizing for an objective given a fixed budget, where the objective include balancing between goal-conditioned importance and
pairwise diversity. In the selection process, the author uses a greedy algorithm, introduces an action-centered AXTree pruning to remove redundant context and accelerate training, and synthesizes reasoning traces. Experimental results demonstrate that the authors' method achieved SOTA performances on three benchmarks, where the authors finetuned three models on subsets of AgentTrek and/or NNetNav that are selected through their pipeline.

**Compliance With Llm Reviewing Policy:**

Affirmed.

**Final Justification:**

My main concerns were addressed so I increased the scores.

**Key Questions For Authors:**

See Summary of Weaknesses.

**Limitations:**

Yes

**Strengths And Weaknesses:**

Summary of Strengths:
- The motivation of the paper is well stated and grounded. Not all steps of every trajectory are useful towards training, and the paper effectively pointed this out and exploited this for more effective and efficient web agent training.
- The method could be easily plugged in to existing agent training datasets.
- The paper attempts to balance the tradeoff between importance and diversity.

Summary of Weaknesses:
- The author's proposed pipeline contains too many elements (e.g. importance-diversity data selection, AXTree pruning, and reasoning generation). It's unclear to me which part contributes the most to the authors' results.
- The authors didn't report the cost of the pipeline. The gain of higher performances and speedup comes at a cost of prices for the selection process, and I assume synthesizing reasoning trajectories might be expensive. Therefore, the authors should report the cost too, allowing the readers to better understand the tradeoffs.

---

> ### Author Rebuttal · Authors · 2026-03-31
>
> Thank you for the positive feedback. We address your questions and concerns below.
>
> ---
>
> **Q1. The author's proposed pipeline contains too many elements (e.g. importance-diversity data selection, AXTree pruning, and reasoning generation). It's unclear to me which part contributes the most to the authors' results. (Reviewers 6iYw, 7RVZ, BwFe)**
>
>
> We provide an ablation study of the key components of our method on WebArena-Lite using Qwen3-8B in the below table. Specifically, we isolate the effects of data selection and reasoning synthesis. The results show that each component contributes meaningful gains, and that combining both yields the best performance, indicating that both are important to the overall effectiveness of WEASEL.
>
>
> | Method | Data Selection | Reasoning Synthesis | WebArena-Lite |
> |---|---|---|---|
> | Qwen3-8B | – | – | 13.5 |
> | SFT (Random) | ✗ | ✗ | 15.8 |
> | SFT (Random + RS) | ✗ | ✓ | 17.6 |
> | WEASEL (Original reasoning) | ✓ | ✗ | 18.2 |
> | WEASEL (Ours) | ✓ | ✓ | **20.0** |
>
> *Table1. Ablation of data selection and reasoning synthesis on WebArena-Lite.*
>
> For target-centered AXTree pruning, our goal is to improve training efficiency and alleviate context-length limitations. As shown in Table 1 of the main paper, this pruning improves training speed by up to 2.2x  while incurring a slight performance tradeoff. In Table 3 of the paper, we further show that it outperforms other AXTree pruning methods, confirming that our pruning is not only efficient but also more effective than other alternatives.
>
> ---
>
> **Q2. The authors didn't report the cost of the pipeline. The gain of higher performances and speedup comes at a cost of prices for the selection process, and I assume synthesizing reasoning trajectories might be expensive. Therefore, the authors should report the cost too, allowing the readers to better understand the tradeoffs.**
>
> We thank you for this suggestion. We note that the WEASEL pipeline incurs a one-time preprocessing cost during training, after which the selected training set can be reused for multiple training runs without additional overhead. On the full AgentTrek dataset, the total preprocessing time is 7.7 hours with 4 RTX 6000 GPUs, including importance-diversity computation, greedy subset selection, and reasoning synthesis. This cost is substantially smaller than the time required to train on the full dataset (136 hours with 4 RTX 6000 GPUs), while enabling the training speedups reported in the paper. We will include this cost breakdown in the revised paper.
>
>
> | Stage | GPU Hours | GPU Type | #GPUs |
> |---|---|---|---|
> | Importance Computation | 0.35 h | RTX6000 | 1 |
> | Diversity Computation | 0.50 h | RTX6000 | 1 |
> | Greedy algorithm | < 0.02 h | - | - |
> | Reasoning synthesis | 7.5 h | RTX6000 | 4 |
>
> *Table 2: Cost breakdown of the WEASEL preprocessing pipeline on the full AgentTrek dataset.*

---

> > ### Author Rebuttal · Reviewer_6iYw · 2026-04-01
> >
> > Thank you for the response! I will increase the overall score to 5.

---

> > > ### Author Response · Authors · 2026-04-02
> > >
> > > Thank you for the response and for updating the score. We sincerely appreciate it.

---

### Official Review · Reviewer_w9xm · 2026-03-13

**Soundness:** 2
**Presentation:** 3
**Significance:** 3
**Originality:** 3
**Overall Recommendation:** 4
**Confidence:** 3

**Summary:**

This paper introduces WEASEL, a trajectory selection method for training web agents that addresses out-of-domain generalization. The approach selects a fixed-budget subset of trajectory steps from offline demonstrations by balancing unary importance with pairwise diversity. The authors also propose action-centered AXTree pruning to keep only content around the ground-truth action target. Experiments show training speed-ups and improved zero-shot transfer across web benchmarks.

**Compliance With Llm Reviewing Policy:**

Affirmed.

**Final Justification:**

After reviewing the authors' rebuttal, I have raised my score from 3 to 4. Below, I explain how the rebuttal addressed my main concerns.

---

Response to Q1 (Importance Score): The authors provided a reasonable defense by (1) pointing to empirical evidence, (2) noting that the diversity term helps mitigate redundant selections, and (3) conducting new experiments with Qwen3-Embedding to show the method's effectiveness is not tied to a specific similarity function. However, I note that BERTScore slightly outperforms Qwen3-Embedding (13.3 vs. 12.7), which is somewhat counterintuitive if deeper semantic understanding were better. While not fully conclusive, the response adequately alleviates my concern.

Response to Q2 (Greedy Approximation Quality): The authors provided excellent empirical validation, showing that the greedy algorithm finds the exact optimal solution. The stability analysis across three independent fine-tuning runs (13.7 ± 0.69) further demonstrates the robustness of the approach. This response fully addresses my concern.

Overall Assessment: The paper's strengths, i.e., clean formulation, practical target-centered pruning, and thorough ablations, remain compelling. My original concerns about the importance score design and greedy approximation have been substantially addressed, particularly the latter. While the method still has room for improvement in capturing deeper task-relevant signals, the empirical effectiveness and the authors' thorough rebuttal justify a recommendation.

**Key Questions For Authors:**

See Weaknesses.

------------------------------------

I am open to raising my score if the author could address my concerns.

**Limitations:**

The authors discuss limitations in the Impact Statement, mentioning potential misuse risks.

**Strengths And Weaknesses:**

**Strengths**

- The formulation is clean, which casts trajectory curation as maximizing a modular objective that explicitly trades off importance and diversity through a single hyperparameter is intuitive and principled, with the greedy algorithm providing reasonable scalability.

- Target-centered pruning is a practical contribution that addresses the bloated AXTree problem directly by retaining only local context around the gold action, leading to substantial token reductions without hurting performance.

- The ablations are thorough, systematically isolating the effect of various components, making it clear which components actually matter and confirming that both importance and diversity contribute complementary signals.

**Weaknesses**

- The importance score relies entirely on BERTScore similarity between step content and goal, which is a shallow semantic signal that doesn't capture whether a step is actually useful for learning the task structure. For instance, redundant steps that happen to mention goal keywords could score high while critical reasoning steps score low if they don't lexically overlap with the goal.

- The greedy algorithm's approximation quality isn't analyzed. While the paper cites a 2-approximation result for max-sum diversification, the actual objective in Eq. 1 includes both importance and diversity with a tradeoff parameter, and it's unclear how far the greedy solution is from optimal or whether the selected subsets are stable across runs.

---

> ### Author Rebuttal · Authors · 2026-03-31
>
> Thank you for the detailed feedback. We address your questions and concerns below.
>
> ---
>
> **Q1. The importance score relies entirely on BERTScore similarity between step content and goal, which is a shallow semantic signal that doesn't capture whether a step is actually useful for learning the task structure. For instance, redundant steps that happen to mention goal keywords could score high while critical reasoning steps score low if they don't lexically overlap with the goal.**
>
> Thank you for this thoughtful concern. We would like to clarify that in web agent trajectories, the states critical for task completion tend to contain goal-relevant content in the AXTree. For example, for the goal of "find the cheapest Sephora brush," the key decision-making state contains elements like "Price low to high" that align semantically with the goal (Figure 2). Table 5 in the paper further confirms that the goal-state pairing yields the best performance (13.3%), outperforming goal-reasoning (10.3%) and goal-state summary (11.5%), indicating that raw state content is empirically a reliable proxy for step relevance.
>
> We also note that our objective is not based solely on the importance term and additionally includes a pairwise diversity term over states and reasoning, which helps reduce the selection of redundant steps, even when they contain goal-related keywords.
>
> To address whether the method depends on the lexical overlaps, we further evaluate WEASEL with embedding models that capture deeper semantic relationships: Qwen3-Embedding-4B and Qwen3-Embedding-0.6B, replacing BERTScore for both $\Phi(t)$ and $D(i,j)$:
>
>
>
> | Method | WebArena-Lite |
> |---|---|
> | SFT (Random, 10K) | 8.5 |
> | WEASEL (Qwen3-Embedding-0.6B) | 12.7 |
> | WEASEL (Qwen3-Embedding-4B) | 12.1 |
> | WEASEL (BERTScore; roberta-large) | **13.3** |
>
> *Table1. Effect of similarity model on WebArena-Lite*
>
> All variants consistently outperform the baselines, suggesting that WEASEL's gains stem from the importance-diversity objective itself, not a specific similarity function. We will include these results in the revision.
>
> ---
> **Q2. The greedy algorithm's approximation quality isn't analyzed. While the paper cites a 2-approximation result for max-sum diversification, the actual objective in Eq. 1 includes both importance and diversity with a tradeoff parameter, and it's unclear how far the greedy solution is from optimal or whether the selected subsets are stable across runs.**
>
> Thank you for this important question. We first clarify that the 2-approximation result from [1] does not directly apply to our objective, since the distance D(i,j) (Eq. 3) does not satisfy the metric properties, as explained in Section 2.3.
>
> Nonetheless, the greedy algorithm finds very strong solutions in practice. To validate this, we conduct an analysis on the NNetNav dataset. For each trajectory of length 10 to 37, we enumerate all possible subsets of size $T_0=3$ and compute the exact objective value (Eq. 1) for each, comparing the greedy solution against the global optimum. Across 1,877 trajectories (lengths 10–37, yielding up to 7,770 candidate subsets per trajectory), we find that more than 96% of times the greedy algorithm finds the exact optimal solution, 99.7% of times find top 1% solution of all subsets, and the ratio between the greedy solution and the optimum achieves the average of 0.9999 with the standard deviation of 0.0005. These results demonstrate that despite the lack of formal metric guarantees, the greedy heuristic is near-optimal in practice for our importance-diversity objective.
>
> Regarding the stability across the runs, we note that the greedy algorithm itself has no randomness in initialization or selection, so the output is stable across runs by construction. Additionally, we also evaluate end-to-end stability across 3 independent fine-tuning runs of Qwen2.5-7B-Instruct. WEASEL consistently achieves the strongest performance across runs, indicating that its gains are stable in practice and not due to a single favorable run.
>
>
>
> | Method | WebArena-Lite |
> |---|---|
> | Qwen2.5-7B-Instruct | 5.9 ± 0.69 |
> | Pruning + Sampling (10K steps) | 8.7 ± 0.35 |
> | Pruning + LLM-Judge (10K steps) | 6.9 ± 1.50 |
> | WEASEL (10K steps) | **13.7 ± 0.69** |
>
> *Table2. WebArena-Lite performance over 3 independent fine-tuning runs of Qwen2.5-7B-Instruct.*
>
> ---
> **References**
>
> *[1] Borodin, A., Jain, A., Lee, H. C., and Ye, Y. Max-sum diversification, monotone submodular functions, and dynamic updates. ACM Transactions on Algorithms (TALG), 13 (3):1–25, 2017.*

---

> > ### Author Rebuttal · Reviewer_w9xm · 2026-04-04
> >
> > Thanks for the detailed response, which addresses my concern through further analysis. I will increase my score to 4.

---

> > > ### Author Response · Authors · 2026-04-04
> > >
> > > Thank you for the response and for updating the score. We sincerely appreciate it.

---

### Decision · Program_Chairs · 2026-04-30

**Decision:**

Accept (regular)

**Comment:**

Reviews ended up quite positive for this work. The main concerns, including reliance on BERTScore similarity, requesting ablation studies to understand the necessity of each proposed component, and comparison to simpler baselines, were adequately addressed during the rebuttal phase. I encourage the authors to incorporate these experiments and discussions into the final revision.